# NOVAsort for error-free droplet microfluidics

Han Zhang [1,8], Rohit Gupte [2,8], Yuwen Li[1], Can Huang [1], Adrian R. Guzman [1], Jeong Jae Han[3], Haemin Jung [1], Rushant Sabnis[4], Paul de Figueiredo[5,6,7] & Arum Han [1,2,4] ✉

High-throughput screening techniques are pivotal to unlocking the mysteries of biology. Yet, the promise of droplet microfluidics in enabling single-cell resolution, ultra-high-throughput screening remains largely unfulfilled. Droplet sorting errors caused by polydisperse droplet sizes that are often inevitable in multi-step assays have severely limited the effectiveness and utility of this technique, especially when screening large libraries. Even a relatively low 1% sorting error results in 10,000 false calls in a 1,000,000 droplet screen, imposing an unreasonably large burden on downstream validation. Here, we present NOVAsort (Next-generation Opto-Volume-based Accurate droplet sorter), a device capable of discerning droplets based on both size and fluorescence intensity. With a 1000- and 10,000-fold reduction in false positives and false negatives, respectively. NOVAsort addresses the challenges of conventional droplet sorting approaches and sets standards for accuracy and throughput in droplet microfluidic assays.

In recent years, droplet microfluidics-based screening systems have proven useful in many different biotechnology and biomedical applications, including discovering rare cell phenotypes[1–3], performing functional interrogation of single cells[2,4–6], discovering microbial pathogens[7–9], or probing the outcomes of microbial community interactions[10,11]. A unique advantage of droplet microfluidics-based approaches is that a large and complex library can be rapidly screened at single-cell and/or single-molecule resolution. Significant technological strides have been made in the last decade in almost all droplet manipulation steps, including droplet transport[12,13], droplet merging[14,15], droplet sorting[16–18], and other droplet manipulation steps. Of all these steps, droplet sorting is often considered the most critical component in screening campaigns[19], as errors in this final step can cause false-positive and/or false-negative results.

Droplet sorting can be performed using pneumatic[20], magnetic[21], thermal[22], acoustic[22], and electric[14,15,19,23,24] methods. The most broadly adopted droplet sorting strategy has been dielectrophoresis (DEP)-based sorting, where a droplet is deflected using a force generated by a non-homogeneous electric field generated within a microfluidic channel[17]. DEP-based droplet sorter designs are simple and easy to fabricate[25], where either surface micromachined electrodes placed at the bottom of a channel or liquid metal- or saltwater-filled microchannels that form three-dimensional (3D) electrodes are used[26]. In the latter, the electrode design can be easily changed to increase the time over which the DEP force acts and thus achieve a larger displacement at the same droplet flow speed and voltage conditions compared to surface electrode-based DEP droplet sorters[17]. Several recent advancements have been made in this area, such as biasing false-positive (FP) droplets to a waste channel to improve sorting accuracy, and preventing drop splitting at the sorter exit junction by a gapped divider[16]. Despite these advancements, further increasing the throughput of droplet sorters becomes challenging, as larger droplets

[1]Department of Electrical and Computer Engineering, Texas A&M University, College Station, Texas 77843, USA. [2]Department of Biomedical Engineering, Texas A&M University, College Station, TX 77843, USA. [3]Department of Multidisciplinary Engineering, Texas A&M University, College Station, TX 77843, USA. [4]Department of Chemical Engineering, Texas A&M University, College Station, TX 77843, USA. [5]Christopher S. Bond Life Science Center, University of Missouri, Columbia, MO 65211, USA. [6]Department of Molecular Microbiology and Immunology, School of Medicine, University of Missouri, Columbia, MO 65212, USA. [7]Department of Veterinary Pathobiology, University of Missouri, Columbia, MO 65211, USA. [8]These authors contributed equally: Han Zhang, Rohit Gupte. ✉e-mail: arum.han@ece.tamu.edu

break into smaller droplets more easily at high flow speed due to the increase in shear stress, combined with sudden changes in momentum when a DEP force is applied to droplets. At the same time, relatively large droplets are often needed in many cell-based assays to support cultivation of cells. To overcome this, a sequentially addressable DEP array (SADA) chip was introduced to enable high-throughput sorting of large droplets without tearing the droplets based on multiple gentle pulls of target droplets. However, the tolerance of this method to polydisperse droplets, which is inevitable in many droplet microfluidics assays, is limited since the flow speed of a droplet is size-dependent[27].

Despite outperforming most other types of sorters, DEP-based droplet sorting methods still suffer from several disadvantages. First, sorting errors arise because of the presence of polydisperse droplets. In most droplet operations, including droplet sorting, highly monodispersed droplets are required to have high accuracy, as each of these functions are designed and optimized for operation with droplets of a specific size range. Thus, the performances of most droplet sorters correlate to the size uniformity of input droplets. However, occasional unintentional droplet merging (resulting in larger droplets) and/or droplet splitting (resulting in smaller droplets) is unavoidable in even the most advanced and well-characterized droplet manipulation steps. This is especially the case during long-term and/or elevated temperature droplet incubation steps that are commonly required for in-droplet cell and molecular biology assay steps. Unintentional droplet merging leads to larger droplets that may contain more than one cell or additional amounts of reagents, and unintentional droplet shredding leads to smaller droplets, both of which lead to false positive or false negative droplet classification and sorting. Since all conventional droplet sorters are based solely on a single droplet parameter such as fluorescent, colorimetric, or impedance signals, incorrectly sized droplets in the absence of droplet size information can easily be sorted as false positives (FP) or false negatives (FN).

Second, high electric field that is not precisely focused on the droplet sorting region can result in unwanted double-sorting. Electrodes of DEP-based droplet sorting systems are often made by filling a microfluidic channel with saltwater or liquid metal[16,17,28]. Unfortunately, the electric field generated by these types of 3D electrodes is not sufficiently localized within the droplet sorting region and can unintentionally actuate multiple adjacent droplets. The large footprint of these electrodes also leads to high electric field strength in the center of the channel, which increases the probability of two adjacent droplets being merged unintentionally. To avoid sorting multiple droplets that are close together, spacing oil is commonly injected to further separate consecutive droplets, which allows only a single droplet to enter the active droplet sorting region at any given point in time[29]. However, stably maintaining even spacing between droplets can be difficult over long periods of operation, especially when the sizes of incoming droplets are not uniform, since smaller droplets tend to catch up with larger droplets due to differences in momentum and drag forces. Having a large droplet-to-droplet spacing also reduces the overall system throughput.

Finally, a higher voltage is needed to achieve higher throughput. Since sorting efficiency is determined by the maximum droplet displacement achievable under a given DEP force, which is dependent on the magnitude and time over which the DEP force acts on each droplet, a higher DEP force is required to redirect a droplet to a different outlet. However, at high voltages (typical operating voltage of existing droplet sorters ranges from 200 to over 1000 $V_{pp}$[16,17,19]), increased inertia of droplets compared to the surface free energy of droplets that maintain their integrity when a displacement force is applied to droplets can result in the droplet tearing apart[16]. This unwanted droplet splitting can severely confound the results of any droplet assay and also limits the assay throughput, especially when relatively large droplets (over 100 μm in diameter) are used, where such larger droplets

are typically more difficult to manipulate but yet are required in many cell assay applications.

We describe herein the NOVAsort (Next-generation Opto-Volume based Accurate droplet sorter) system, which can sort droplets not only based on their fluorescence intensity but also based on their size. Accurately sorting polydisperse droplet populations can significantly reduce or eliminate sorting errors stemming from unintentional effects such as droplet splitting and/or droplet merging even under high-throughput conditions. The presented droplet sorter, which effectively addresses the aforementioned drawbacks and limitations, utilizes interdigitated electrodes (IDEs) that can generate a highly localized electric field at the bottom of the microfluidic channel and also leverages the natural buoyancy of aqueous droplets in carrier oil to selectively manipulate droplets within a specific size range[30]. A flow-through droplet fluorescence detection setup coupled with a feedback loop activates the IDEs only when the droplet in the detection region has an intensity over a set fluorescent threshold, thus only collecting high-fluorescent droplets of the correct size since the DEP force generated by the surface IDEs can only manipulate droplets within a specific size range. Also, the highly localized electric field generated by the planar IDEs moves only the droplets located directly above the IDEs and thus does not coalesce adjacent droplets or sort non-target ones. Moreover, activated IDEs serve as "droplet guide rails" that gently guide the droplets towards the correct outlet as opposed to conventional droplet manipulators that abruptly pull the droplet to a different direction at a sharp angle, which creates a large sudden change in momentum of the droplets that results in unwanted droplet shearing. Finally, as a proof-of-concept demonstration of the advantage of this NOVAsort system, an in-droplet in vitro transcription/translation (IVTT) workflow was performed and the resulting error rates as well as throughputs were thoroughly characterized.

## Results
### Working principle of the NOVAsort system
The uniformity of droplet size can be significantly impacted after multiple steps of droplet incubation and/or manipulation operations. Figure 1A illustrates the typical input droplet composition of a droplet library to be sorted[17]. Here, droplets consist of small (disrupted), medium (target size), and large (unintentionally coalesced) droplets, with or without fluorescence. However, only the medium-sized fluorescent-positive droplets are the true hits that should be sorted. The NOVAsort is designed to deal with this highly polydisperse input droplet library. This is achieved by coupling sized-based droplet separation with fluorescence-activated droplet sorting (Fig. 1B). The procedure of NOVAsort is illustrated in Fig. 1C. The size-based droplet separation method is based on the fact that aqueous droplets are naturally buoyant in their carrier oil and float to the top of the channel and that IDEs located at the bottom of a channel generate a very localized electric field at the channel floor. Large droplets that fill the channel height can be easily manipulated by dielectrophoretic (DEP) force while smaller droplets that float up and thus away from the surface IDEs cannot be manipulated due to insufficient DEP force acting on those smaller droplets (Fig. 1D)[30]. In this work, the channel height is designed to be slightly larger (~10%) than the desired droplet size so that the IDE-generated local electric field specifically influences and guides only droplets with the correct size. By sequentially linking two such size-based droplet manipulators (Fig. 2A shows a droplet high-pass and a low-pass filter connected in series), sorting out only a certain droplet size range becomes possible. The fluorescence detection laser spot is placed at the beginning of the second pair of IDEs, which generate the DEP force to selectively sort only the fluorescent-positive and correct-size droplets.

The design of the overall NOVAsort system is shown in Fig. 2A, using the case of target droplet size of 95 μm in diameter, smaller unwanted droplets of 50 μm (simulating disrupted droplets), and

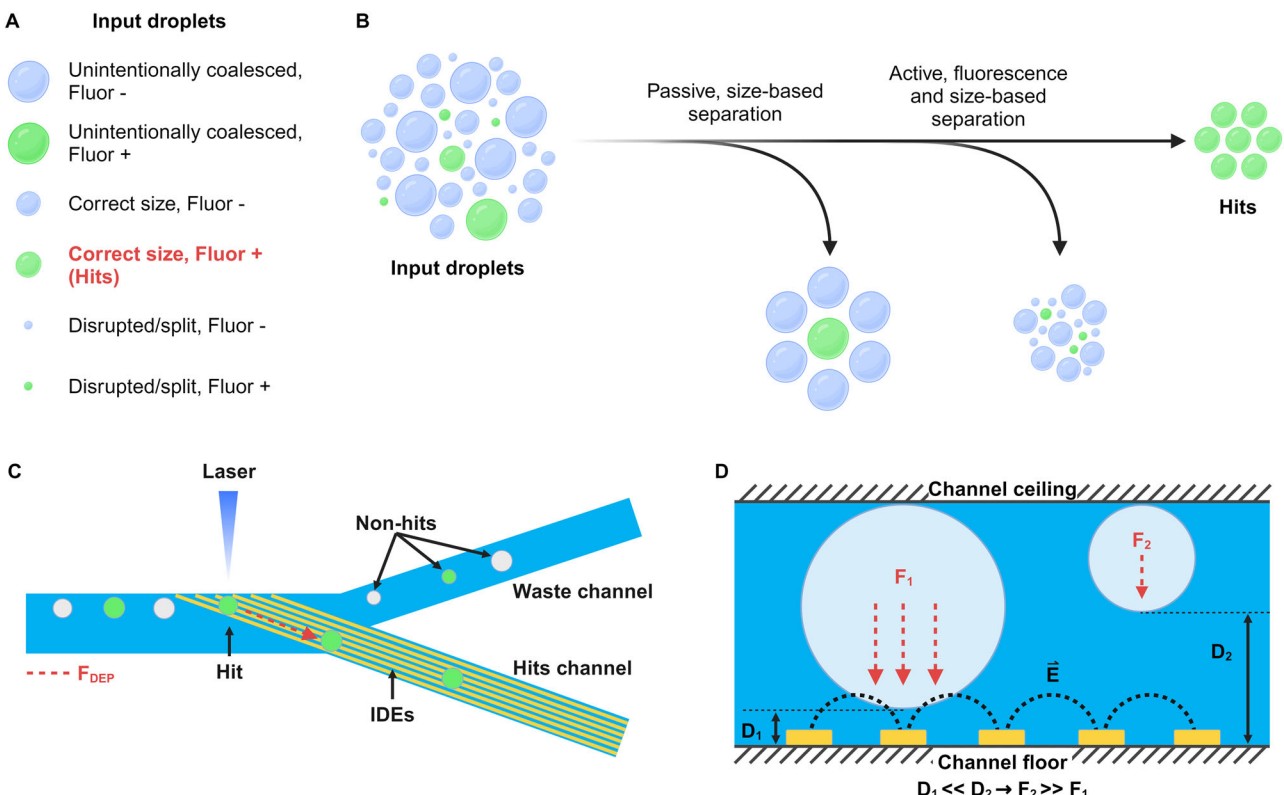

**Fig. 1 | Working principle of the NOVAsort droplet sorter that uses a planar-IDE-generated DEP force and droplet buoyancy for simultaneous size- and fluorescence-based droplet sorting. A** Typical polydisperse droplet library after multiple incubation and/or manipulation steps, which flows into the droplet sorter. **B** Overview of the NOVAsort droplet sorting workflow. **C** Top view of the droplet manipulation region illustrates that larger droplets will be affected by the DEP force and follow the trajectory of the IDE patterns while the smaller droplets will continue to flow uninterrupted to the waste outlet. **D** A cross-sectional view of the IDE region illustrates that a large droplet is closer to the IDE surface and thus experiences a much stronger DEP force ($F_1$), while a small droplet that is buoyant and further away from the IDE surface experiences a much weaker DEP force ($F_2$). Here $E^-$ stands for electrical field, and $D_1/D_2$ stands for the distance between the IDEs and droplets. Figure 1/panels (**A**–**D**), created with BioRender.com, released under a Creative Commons Attribution-NonCommercial-NoDerivs 4.0 International license.

larger unwanted droplet size of 120 µm (simulating merged droplets) as an example case. In Section 1, the droplet spacing and lifting region, the height of the two side spacer oil channel (H: 50 µm) was designed to be half of the main droplet reflow channel (H: 100 µm). The height of the reflow channel was then increased to 1.2 times (120 µm) starting from where the spacer oil flows in. As opposed to conventional oil spacing side channel designs that have equal height as the main channel, NOVAsort creates an oil sheath flow underneath the reflowed droplets, where this sheath flow lifts the aqueous droplets to the ceiling of the channel before they enter the first IDE active region (Section 2), while also spacing them out. In addition to the natural buoyancy force of droplets that lift them up, this design speeds up the droplet lifting process, enabling higher speed operation of the system. Section 2, the large droplet removal region, contains the first IDE array that behaves as a low-pass filter, removing large droplets (>100 µm) from the main channel into waste 1 outlet. The IDE array used here has 5 µm finger width and 5 µm spacing between fingers. In Section 3, the side focusing and droplet pushdown region, the height of the main channel is decreased from 120 to 100 µm before the droplets enter the subsequent sorting junction (see Fig. 2A side view), while all droplets are also biased towards the waste channel side when the electrical field is off (see Fig. 2A top view). The reason to push down all the droplets is so that further size-based separation between small and medium-sized droplets is possible in the subsequent region. In the final Section 4, the target droplet sorting region, the laser spot is placed right in before the second IDE for fluorescent signal excitation. When the emission readout is over a set threshold, the second IDE is activated for a short period (typically milliseconds) to selectively pull the fluorescent

droplet into the hit outlet. Thus, this design allows pulling out all the fluorescent droplets within the desired size range into the "hit" channel, in this case, 80–100 µm diameter (100 µm height channel allows pulling 80+ µm droplets), while allowing the smaller droplets to move undisturbed and into waste 2 outlet as they are unaffected by the DEP force generated by this second IDE. The target droplet size to be sorted can be simply adjusted by changing the channel height of each section as well as the height difference before and after Section 3. The optics, electronics, and software interfaces used in this process are illustrated in Supplementary Fig. 1.

## Proof-of-concept validation

A droplet population consisting of three different sizes (50, 95, and 120 µm in diameter) was used as the input sample. In each droplet size, around 5% of droplets contained a mixture of fluorescent dye and black color dye for fluorescence-activated droplet sorting while also being able to easily visualize and quantify the sorting results, respectively. The input droplet components used here are illustrated in Fig. 2D. The device was designed to have 120 µm height channel in Section 3 so that all 120-µm-diameter droplets could be removed (Fig. 2C left and Supplementary Movie 1). Section 4 channel height was 100 µm so that all 50 µm droplets, even if they are fluorescence-positive, could be moved into waste 2 outlet (Fig. 2C right). Blank 95 µm droplets were also removed into waste 2 outlet since they are not fluorescent, and thus the sorter was not activated. Thus, only 95-µm-diameter fluorescent droplets were sorted into the "hit" outlet (Fig. 2C middle). More than 1000 droplets going through the droplet sorting system were tracked and quantified (Fig. 2D). The error rate for waste 1

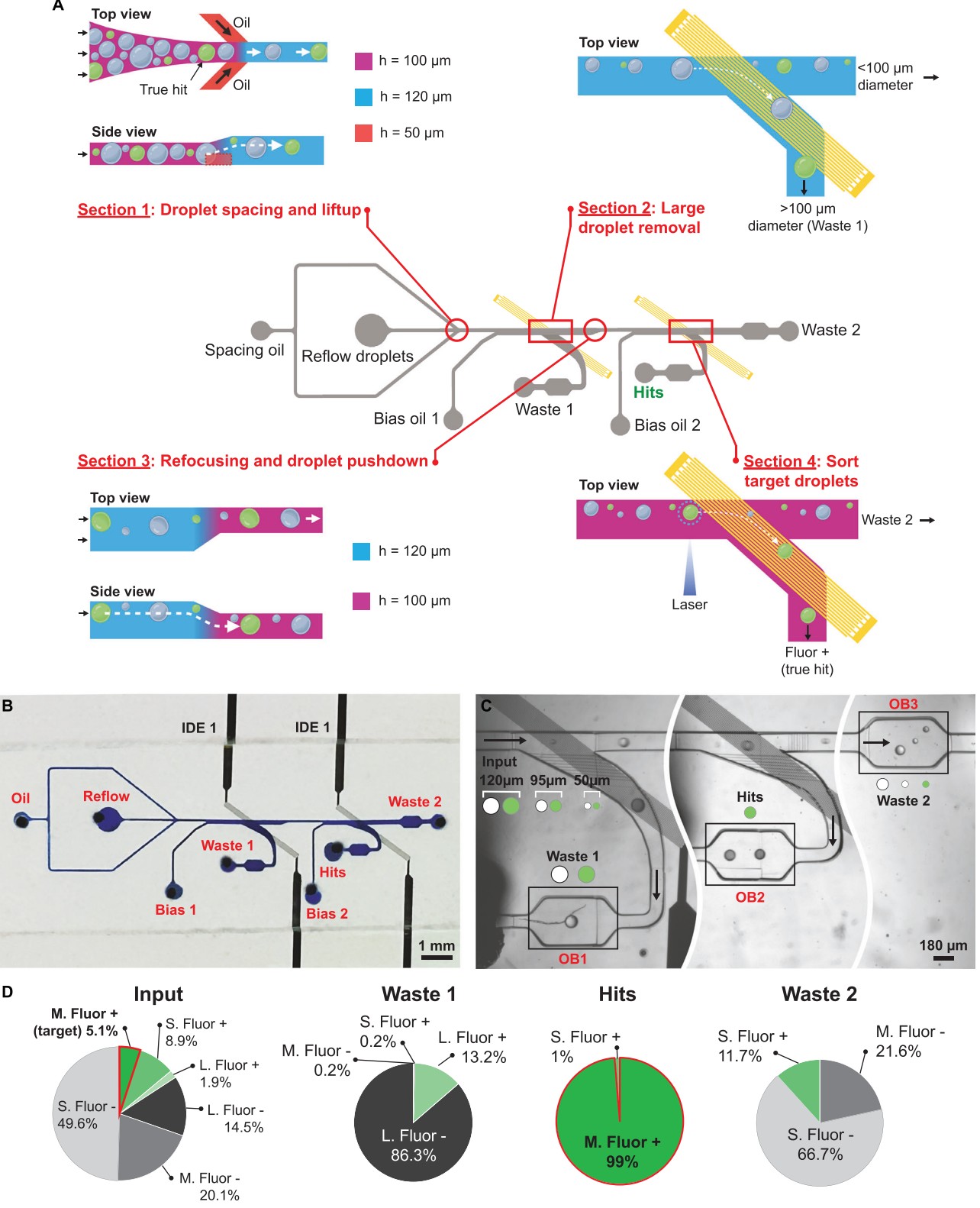

**Fig. 2 | Design of NOVAsort and proof-of-concept. A** NOVAsort design shows the four functional sections of the sorter, each with an enlarged view (Section 1 "droplet spacing and liftup"; Section 2 "large droplet removal"; Section 3 "refocusing and droplet pushdown"; Section 4 "droplet sorting"). **B** A micrograph of the fabricated device with blue color dye filling the channels for easy visualization. **C** Representative micrograph (three separate experiments with similar results) of a six-size input droplet sorting demo. Large droplets are sorted to the waste 1 outlet, medium-sized fluorescent droplets are sorted into the "hits" channel, and the smaller droplets are sorted into the waste 2 outlet. OB=observation chamber. **D** Analysis of the experiment performed in (**C**). The composition of droplets in the input, waste 1 outlet (453 droplets), hit outlet (108 droplets), and waste 2 outlet (1589 droplets). L, M, S indicate large, medium, and small droplets; Fluor+ and Fluor- indicate fluorescence-positive and fluorescence-negative droplets. Part of **A**, created with BioRender.com, released under a Creative Commons Attribution-NonCommercial-NoDerivs 4.0 International license.

outlet was 0.4% ($n = 453$, error defined as droplets that are not large size), hit outlet was 0.9% (error defined as droplets that are incorrect size or fluorescent negative), and waste 2 outlet was 0% (error defined as droplets that are correct size and fluorescent positive). Overall, the operation accuracy of the system was greater than 99% at a throughput of 30 Hz. A video demonstration of the entire droplet sorting process is shown in Supplementary Movie 2.

## Performance evaluation of NOVAsort compared to a conventional linear sorter

To evaluate the performance of NOVAsort when the input droplet population is highly polydisperse, 95 μm diameter green fluorescent (mixed with black color dye) droplets were spiked into a sonicated polydisperse clear droplet pool to assess whether these 95 μm diameter fluorescent droplets can be correctly retrieved (Fig. 3A). The NOVAsort was quantitively compared against a commonly used "linear sorter"[16] (linear sorter result: Fig. 3B; NOVAsort result: Fig. 3C). The device design and experimental setup for the linear sorter are shown in Supplementary Fig. 3. In the case of the linear sorter, the collected hits contained a large number of droplets without fluorescent and/or incorrect size. Here, collection from the hit outlet shows that the target-size (95 μm) fluorescent droplets were enriched by only 6.6 fold, from 3.6 to 23.6% (Fig. 3D). In contrast, when using the NOVAsort, almost all droplets collected from the hit outlet were both fluorescent positive and correct target size where the target-size (95 μm) fluorescent droplets were enriched from 5.8 to 98.8% (Fig. 3D). Here, the waste 1 outlet had 99.8% accuracy in successfully removing the large droplets (Supplementary Fig. 2). After droplets were processed through sections 2 and 3, the NOVAsort was able to separate the target droplet at a very low false positive rate (1.2%, 5 out of 414 droplets) and false negative rate (0.3%, 18 out of 6,348 droplets). In contrast, the false positive rate of the linear sorter was significantly higher (23.2%, 1729 out of 7456 droplets). Both experiments were conducted at a throughput of 30 Hz. Additional images of droplets collected from all outlets after sorting are shown in Supplementary Fig. 4. A video demonstration of the droplet sorting process under this highly polydisperse input condition is shown in Supplementary Movie 3.

Next, we evaluated how NOVAsort performs when negative sorting is needed. Examples of such assays are when inhibitory compounds are desired to be screened (e.g., droplets containing molecules or cells that inhibit the growth or expression of fluorescent reporter cells, as in anti-microbial and/or anti-virulent compound screening). In the case of monodisperse droplet library and positive sorting (e.g., small number of droplets are fluorescent positives and need to be selected, 5% of the population in our test case), both sorters performed well at a variety of throughputs (11–235 Hz, Fig. 3E, F), with error rates ranging from 0.5 to 1.5% for both methods ($n = 200$). The performance of the control linear sorter was excellent and NOVAsort does not necessarily outperform the linear sorter. However, in the case of negative sorting where most of the droplets are fluorescence positive (in the tested example 95% of the overall population) and only a small number of fluorescence-negative droplets (5%) must be sorted out, the error rates (FP + FN) of NOVAsort and linear sorter were 0.8% and 7.5%, respectively ($n = 500$). Here, in the case of the linear sorter, the incidences of error are mainly caused by missed droplets and accidental pulling of non-fluorescent droplets due to the frequent activation of the electrical field to sort out fluorescence-positive droplets (Fig. 3G, H). An alternative to this method is to perform a two-fluorescent-color sorting, but that adds to the overall system complexity. The operation of the linear sorter and NOVAsort processing a library having 5% and 95% fluorescence-positive droplets, respectively, as well as positive/negative sorting, are shown in Supplementary Movie 4. Overall, NOVAsort provides approximately 10 times better performance compared to the conventional linear sorter in this negative sorting case.

## Failure modes of conventional sorter—unintentional coalescence and false-positives

In the conventional linear sorter, multiple droplets in the sorting junction that are close to each other can be merged or pulled together into the hit outlet (Fig. 4A). This is due to the non-localized, high-strength electric field generated by the liquid metal 3D side electrodes (Fig. 4A). Figure 4B top shows cases of an unintentional merging of two small droplets with a large droplet (gray color), and Fig. 4B bottom shows pulling of two droplets when the sorter is triggered. In contrast, such failures are not observed in the NOVAsort (Fig. 4C), where even in the case of two droplets coming very close to each other into the sorting zone they do not merge (Fig. 4D top) and can still be sorted one at a time (Fig. 4D bottom). This is because the electric field strength generated by a 3D liquid metal electrode is significantly higher (Fig. 4E) compared to that of the electric field generated by surface IDEs that diminishes rapidly outside of the small, localized region near the bottom of the channel (Fig. 4F). Thus, the localized electric field actuates only the droplets present directly above the IDEs, preventing unintentional double sorting, and where merging of even insufficiently spaced droplets is unlikely to occur. Supplementary Fig. 5 compares the top views of the generated electric field in the two different electrode designs.

NOVAsort also showed good tolerance to small droplet-to-droplet distance. Here, a monodisperse emulsion of 80 μm diameter droplets with 5% of the droplets containing fluorescent dye and black ink mixture was flown at different droplet-to-droplet distances (1600 μm, 800 μm, 400 μm, and 200 μm) and the sorting results from the two droplet sorters were compared. As can be seen in Fig. 4G, for the linear sorter the sorting accuracy dropped dramatically from 97.4% at 1600 μm droplet-to-droplet distance to 16.8% at 200 μm droplet-to-droplet distance. The major causes were: (1) pulling of multiple droplets in one trigger, (2) unintentionally merged droplets being incorrectly sorted (hit and blank droplets merged and pulled into the hit outlet), (3) unintentionally merged droplets not being sorted (hit and blank droplets merged and exit through waste outlet). Examples of these failure models are shown in Supplementary Movie 5. In contrast, no significant decrease in sorting accuracy was observed when using NOVAsort even when the droplet-to-droplet distance was reduced (from 100% sorting efficiency at 1600 μm droplet-to-droplet distance to 96.6% at 200 μm droplet-to-droplet distance). No undesired merging was observed even when the droplets were in close contact with one another (Supplementary Movie 6). Pulling multiple droplets at the sorting junction was observed only at a very small droplet-to-droplet distance (200 μm), but very rarely. NOVAsort sorting at different droplet-to-droplet distances can be seen in Supplementary Movie 3.

To determine the minimum droplet-to-droplet distance needed to prevent false-positive sorting, the droplet trajectories under a variety of different electric field activation conditions were tracked (Fig. 4H and Supplementary Fig. 6). The minimum sorting delay time for the IDE to unsuccessfully sort a target droplet was 4 ms. At the droplet throughput of 40 Hz, which translates to 43 μm·ms⁻¹ (Supplementary Fig. 7), this means that the IDE sorter does not incorrectly sort close-by droplets if center-to-center distances between the droplets are more than 172 μm. Frame-by-frame micrograph images of droplet sorting at different conditions described in Fig. 4H and Supplementary Fig. 5 are shown in Supplementary Fig. 8. In addition, the successful sorting of a droplet into the hit outlet requires the DEP force to pull the center of the droplet below the critical border separating the two outlets. The position of this border is determined by the flow rate ratio between the hit outlet and the waste 2 outlet. The shift of this critical border under different input flow rate conditions is shown in Supplementary Fig. 9. A low flow rate ratio (flow rates of outlet 2 vs. outlet 3 < 0.35) or high flow rate

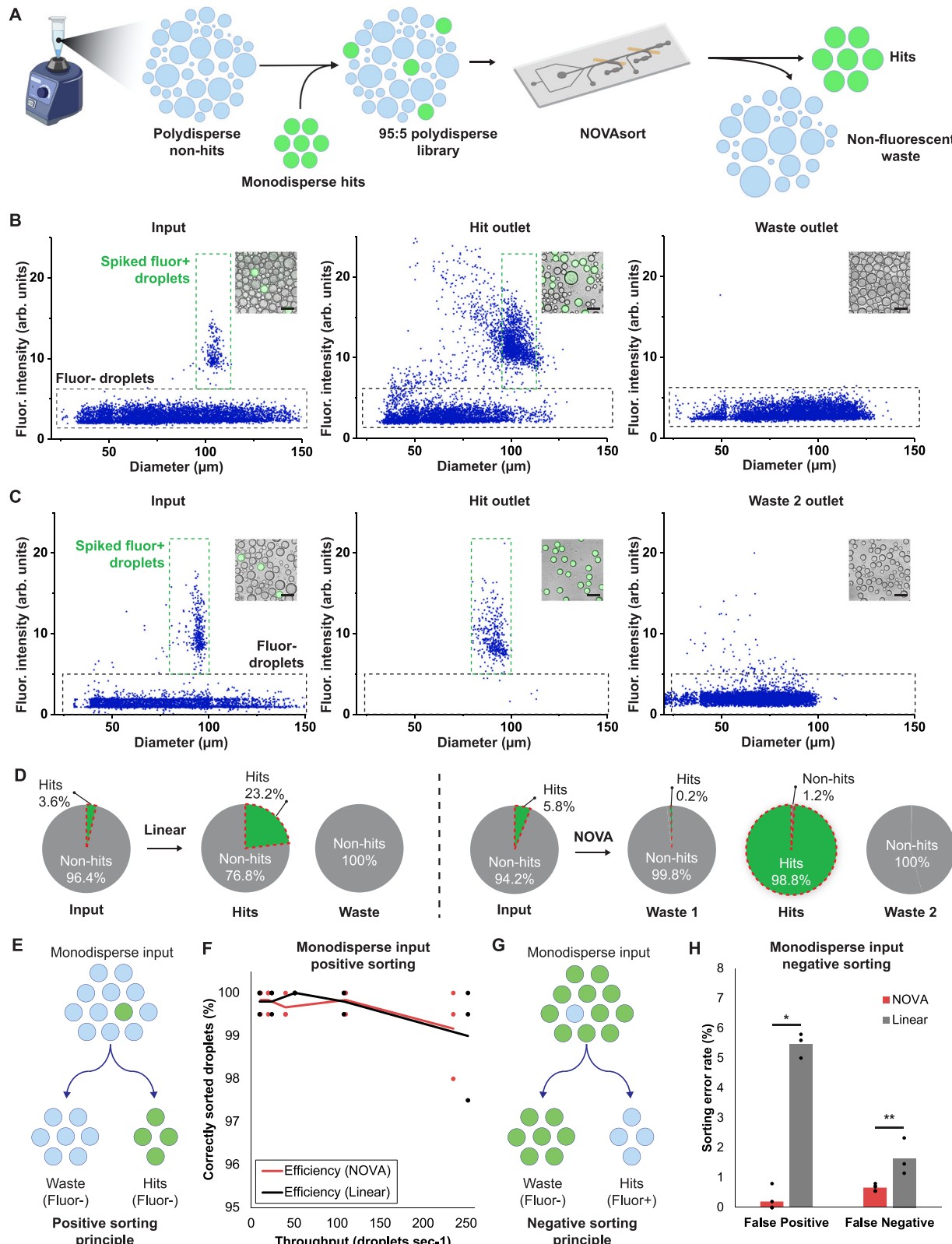

**Fig. 3 | Performance comparison of the NOVAsort to the conventional linear sorter using mono- and poly-disperse droplet libraries. A** Procedure to recover the spiked-in fluorescence-positive "hit" droplets from a polydisperse droplet library. Size distribution and fluorescent intensity of input droplets, droplets coming out of the hit outlet, and droplets coming out of the waste outlets in the case of (**B**) linear sorter and (**C**) NOVAsort (scale bar = 180 μm). **D** Droplet compositions pre- and post-sorting (Linear sorter: input = 1504 droplets; hit = 240 droplets; NOVAsort: input = 1184 droplets; hit = 171 droplets). **E, F** Positive sorting

performance comparison when 5% of the droplets are fluorescence positive (i.e., hits) and must be sorted out (*n* = 3). Data are presented as mean values ± SD. **G, H** Negative sorting performance comparison when 95% of the droplets are fluorescence-positive and where fluorescence-negative droplets (i.e., hits) must be sorted out (*n* = 3). One-Way ANOVA, *: $p = 0.12 \times 10^{-6}$, **: $p = 0.01$. Data are presented as mean values ± SD. **F, H** Each condition has data from three separate experiments. **A**, created with BioRender.com, released under a Creative Commons Attribution-NonCommercial-NoDerivs 4.0 International license.

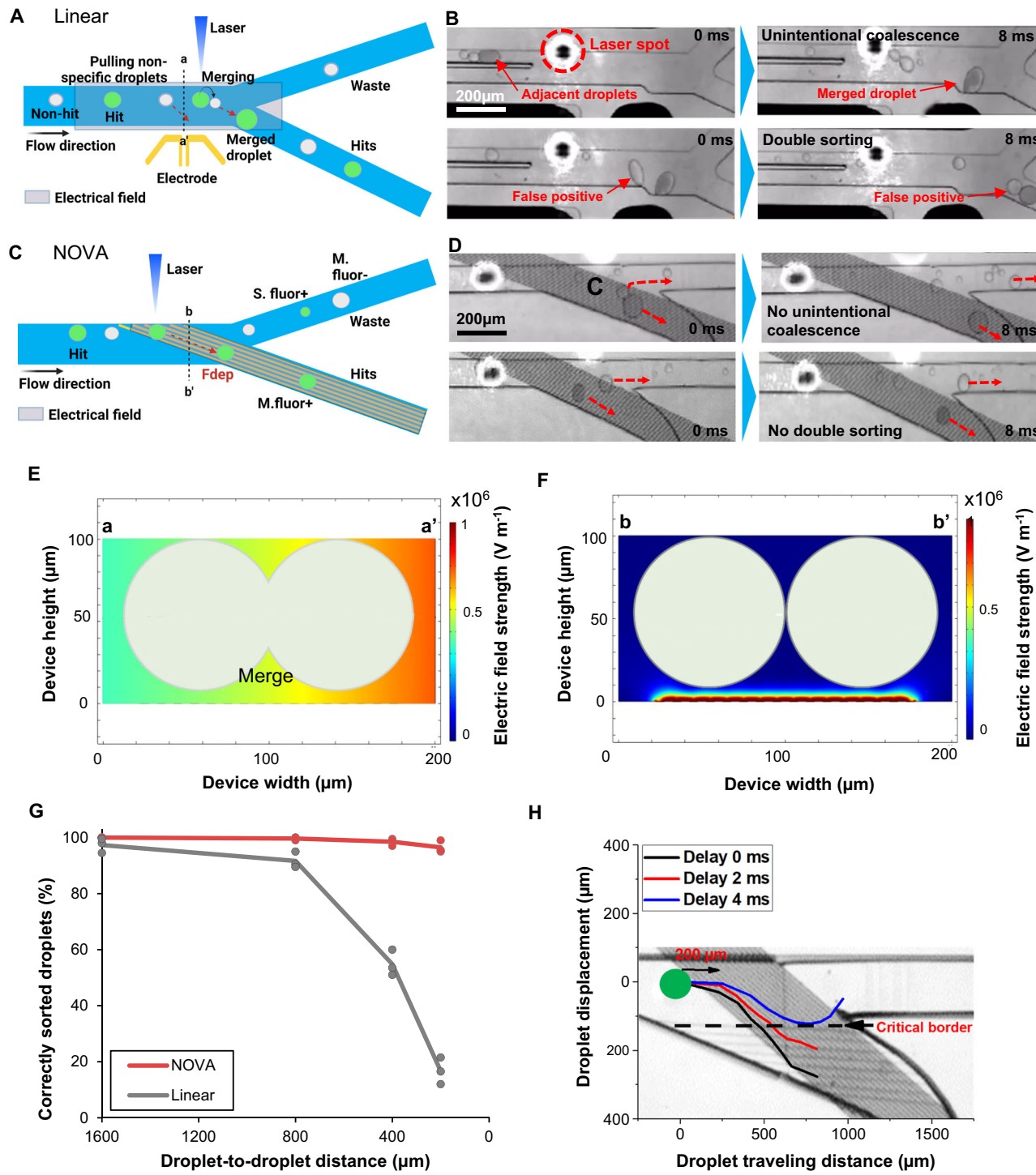

**Fig. 4 | Overcoming unwanted droplet merging and false-positive droplet pulling errors using NOVAsort. A** Schematic and (**B**) micrographs of unintentional droplet merging (top) or multiple droplet sorting (bottom, false positive) when using a linear sorter (scale bar = 200 μm). **C** Schematic and (**D**) micrographs of NOVAsort preventing false-positive sorting (top: blank droplet right behind the larger and gray "hit" droplet correctly going out to waste), showing good tolerance to even close droplet-to-droplet distance due to the localized electric field. Simulated electric field strength generated by the (**E**) liquid metal electrodes (600 V_pp,

10 kHz) and (**F**) surface IDEs (20 V_pp, 10 kHz) in the case of a 100 μm height channel. **G** The accuracy of sorting a correctly sized and fluorescent-positive droplet from a polydisperse library under different droplet-to-droplet distances. Data are presented as mean values ± SD of three separate experiments ($n = 3$). **H** The sorting trajectory of droplets under different sorting delay times. The delay time of 2 ms means the electric field is triggered 2 ms after the droplet fluorescence is detected. **A**, **C**, created with BioRender.com, released under a Creative Commons Attribution-NonCommercial-NoDerivs 4.0 International license.

ratio (flow rates of outlet 2 vs. outlet 3 > 1) are not recommended as the droplet biasing can fail under a low ratio (which results in the border being too low), or a much larger DEP force is needed to sort droplets to outlet 2 when the ratio is too high.

## Failure modes of conventional sorter−unintentional droplet splitting

Increasing droplet flow speed to increase throughput comes with the potential for unwanted droplet splitting, especially when the droplet

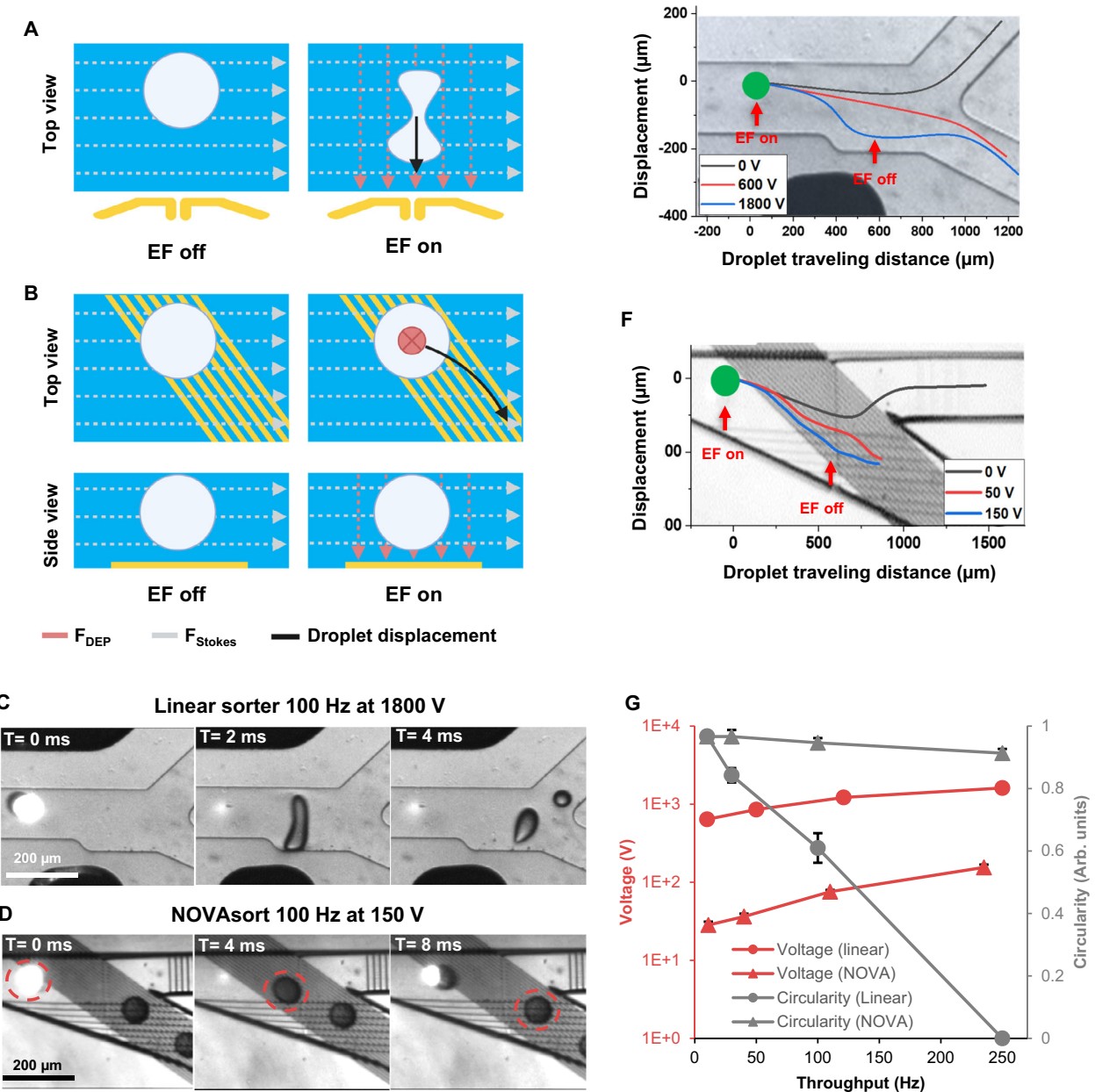

**Fig. 5 | Overcoming unwanted droplet splitting using NOVAsort. A** Schematic of droplet splitting caused by DEP force pulling a droplet from the side of the sorting channel perpendicular to the Stokes force direction in the linear sorter. **B** Schematic showing that the IDE-based NOVAsort can solve the droplet splitting problem by placing a sorting electrode underneath the droplet as guide rails. Frame-by-frame micrographs comparing the sorting of a 110 μm diameter droplet under electric field generated by the (**C**) 3D liquid metal electrode and (**D**) surface IDE. The droplet can be seen torn apart in the case of the linear sorter (**C**), while the droplet was intact when using the IDE sorter (**D**). Sorting trajectory of droplets under no voltage, minimum sorting voltage, and threefold of the minimum sorting voltage in the case of the linear sorter (**E**) and NOVAsort (**F**); n = 12 repeats. **G** Impact of sorting throughput on minimum operational voltage required (black: linear sorter; red: NOVAsort) and circularity (blue curves) of both sorting methods (50 Hz). Circularity = 0 indicates tearing up of droplets. Data are presented as mean values of at least 11 separate measurements ± SD (n = 11). **A, B** created with BioRender.com, released under a Creative Commons Attribution-NonCommercial-NoDerivs 4.0 International license.

size is relatively large (e.g., >100 μm). Such droplet size is needed, for example, in assays that require in-droplet cell cultivation. In the linear sorter, large droplets break into smaller daughter droplets due to insufficient surface tension that holds the droplets together (Fig. 5A). This is because as the flow speed increases, the Stokes force increases, requiring a larger DEP force to pull the droplet. This combination of large DEP force and large Stokes force could overcome the droplet surface tension and tear the droplets. In the case of NOVAsort where surface IDEs are located at the bottom of the flow channel, instead of

pulling droplets from the side of the fluidic channel as is the case in the liner sorter (Fig. 5A), the surface IDEs serve as guide rails that gradually pulls the droplets to follow the IDE patterns. To experimentally demonstrate this, large droplets (110 μm) and high voltage (150 $V_{pp}$ for NOVAsort and 1800 $V_{pp}$ for the linear sorter) were used to observe droplet shredding at a relatively low throughput (100 Hz, 120 μm·ms⁻¹). Figure 5C, D shows frame-by-frame micrograph images that compare the droplet sorting trajectories in the linear sorter (Fig. 5C) and the NOVAsort (Fig. 5D). Contrary to a droplet being torn apart when pulled

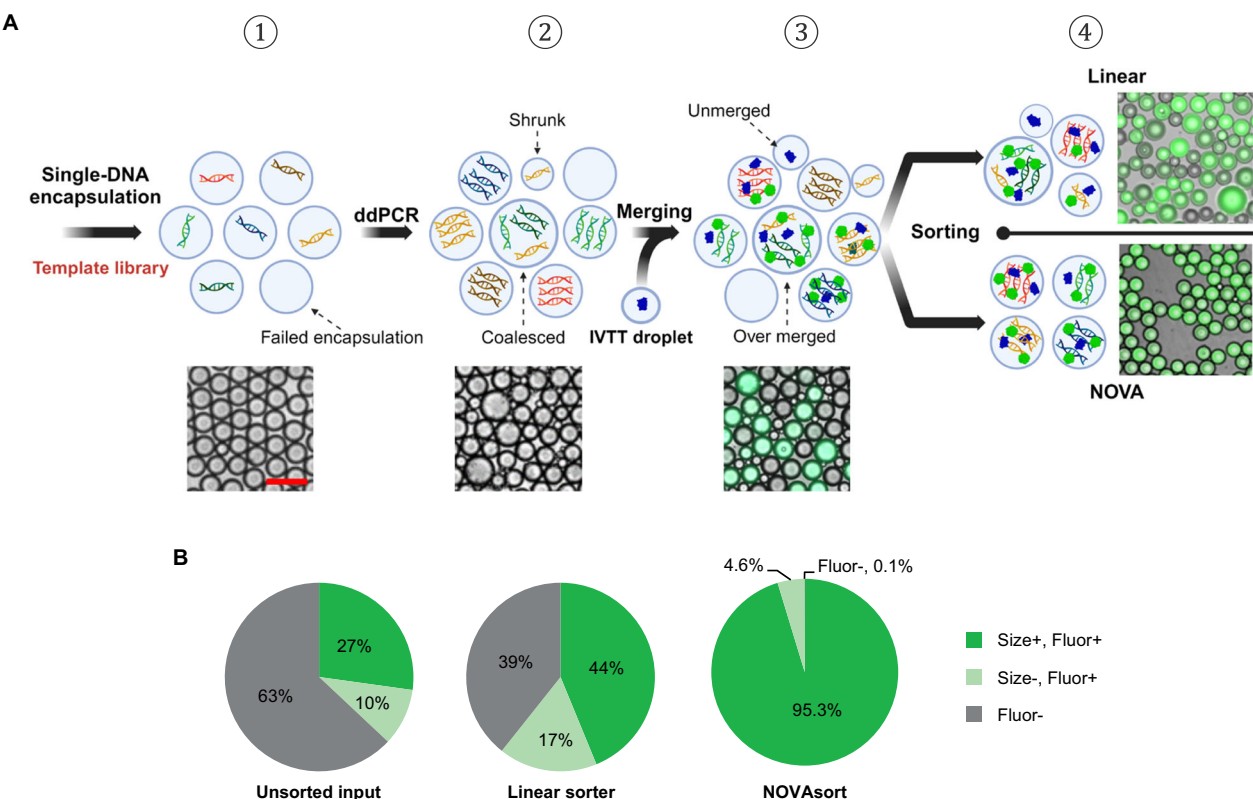

**Fig. 6 | Droplet-based IVTT library generation and sorting using the two different droplet sorters. A** Step-by-step workflow of the droplet-based IVTT assay, where the generated polydisperse library is sorted by either the linear droplet sorter or NOVAsort (scale bar = 200 μm). **B** Sorting results of the polydisperse IVTT droplet library (*n* = 1000 for the "linear sorter" group and *n* = 1000 for the "NOVAsort" group). "Size +" and "Size −" stand for droplets with correct and incorrect sizes. "Fluor +" and "Fluor −" stand for GFP-positive and GFP-negative droplets. "Size+ & Fluor +" are the target hits that need to be sorted. **A** created with BioRender.com, released under a Creative Commons Attribution-NonCommercial-NoDerivs 4.0 International license.

by the DEP force from the side of a fluidic channel using a linear sorter (Supplementary Movie 7), the droplet remains intact in the NOVAsort without being shredded (Supplementary Movie 8). The sorting trajectories (50 Hz and 15 ms) in the linear sorter and NOVAsort are shown in Fig. 5E, F, respectively. When the voltage triples (Fig. 5E, from 600 to 1800 V) the DEP force along with the strong Stokes force (vertical direction) tears the droplet apart after the electric field is triggered. In contrast, the trajectory of droplets on the IDEs at different voltages does not abruptly change. Thus, at the same throughput, higher voltage allowed the droplets to rigidly follow the path of the IDEs before they were released into the hit outlet (Supplementary Fig. 10).

We further investigated the operational voltage needed to achieve successful droplet sorting. The operational voltage needed for NOVAsort ranged from 20 $V_{pp}$ at low throughput (11 Hz) to 150 $V_{pp}$ at high throughput (235 Hz). This voltage need is more than 10 times lower than that for the linear sorter at similar throughputs (Fig. 5G). This low operational voltage further reduces the risk of unwanted droplet splitting or merging. The "droplet circularity", defined as the width of the droplet (X-axis) over the maximum length of the droplet (Y-axis) at the droplet pulling stage (circularity = 0 indicates breaking apart of a droplet), showed a dramatic reduction in the case of linear sorter when the throughput increased (Fig. 5G). However, no significant change in droplet circularity was observed in the case of NOVAsort even at a throughput of up to 235 Hz.

## Application example—improving the screening accuracy of IVTT assays

In vitro transcription/translation (IVTT) allows for rapid and less labor-intensive protein synthesis without the need to consider cellular

metabolism or accumulation of toxic byproducts. It can be used to generate a combinatorial synthetic protein library for functional screening using droplet microfluidics[31,32]. However, an important step of IVTT[32], namely the thermal cycling of droplets, leads to severe disruption of the emulsion even under well-optimized conditions[33]. This resulting polydispersity in turn disrupts the droplet merging step that is required to add the IVTT reagents to the droplets that underwent thermal cycling. Here, unmerged or over-merged droplets will result in false positives or false negatives during droplet content detection and thus need to be removed. Fig. 6A shows a typical droplet-based IVTT workflow, which was also used here. First, single DNA template-encapsulated droplet library is generated, which is highly monodisperse (step 1). Next, droplet thermocycling is conducted for DNA amplification (step 2), at which point many unwanted merging occurs. These non-uniformly sized droplets are then merged with the IVTT substrate-containing droplets, ideally one-by-one (step 3). Here, because of the polydispersity of droplets after thermocycling, many unmerged and/or over-merged droplets emerge. Then, the IVTT protein expression is completed within the droplets, in this case the production of green fluorescence protein (GFP), making the protein library ready for sorting (step 4). As can be seen in the sorted droplet images, a notable improvement in droplet size uniformity and sorting accuracy can be achieved when using NOVAsort compared to the linear sorter. As can be seen in Fig. 6B "unsorted input", the resulting droplet library after step 4 included many droplets without GFP expression and/or incorrect size, with only 27.2% of the droplets being the correct size and GFP positive (the target droplets desired to be sorted). Using the linear sorter, only 43.8% of the sorted droplets were confirmed to have GFP and the correct droplet size, with the rest being

false positives. In contrast, when using NOVAsort, 95.3% of the sorted droplets were GFP positive and correct size (217% improved). Thus, the developed NOVAsort could indeed substantially improve the accuracy of droplet microfluidics-based IVTT assays. Detailed images of key steps of this assay are shown in Supplementary Fig. 11.

## Discussion

The polydispersity of droplet libraries dramatically reduces the accuracy of most currently available droplet sorters. To overcome this challenge, we developed the NOVAsort that can selectively sort target droplets that are both correct size and fluorescent positive from a polydisperse library with high accuracy (>99%) at a throughput of up to 235 Hz. The highest throughput reported for a 100-diameter droplet is 1000–2000 Hz[27]. In our work, the rate 235 Hz was chosen based on the maximum frame rate (500 frames/sec) of our high-speed camera (Hamamatsu C1440) to ensure error-free analysis of individual droplets. However, the throughput limit of the fluidic system can be much higher than 235 Hz. Having said that, even at the currently tested throughput of 235 Hz, >800,000 droplets can be processed in less than one hour, which is adequate for most biological applications. NOVAsort also outperforms the conventional linear sorter in the case of negative sorting (99.2% sorting accuracy of NOVAsort vs. 92.5% sorting accuracy of linear sorter), where a large percentage of droplets being fluorescent positive requires frequent triggering of the electric field that typically results in false-positive sorting (pulling more than one droplet at a time per trigger) and unwanted droplet merging. The limited negative sorting accuracy is one of the most common reasons why negative sorting is not often conducted in droplet assays even though such assays are needed in many applications, such as antibody/immunocyte discovery, pathogenicity assays, and cell toxicity screening applications, to name a few.

The NOVAsort accuracy was generally >99%. That is, in a scenario where a library containing 1 million cells must be screened, a 1% error rate means that there will be only 10,000 incorrectly sorted droplets. In contrast, a platform that runs at 80% accuracy will result in 200,000 incorrectly sorted droplets, which is a very large number and can severely impact the downstream analysis, considering that most assays require time-consuming confirmation assays. Thus, NOVAsort technique can improve the efficiency of many different assays that encounter droplet size uniformity issues, such as those that generally require long-term cell cultivation, elevated-temperature incubation, or multiple droplet manipulation steps, commonly needed in single cell-based analysis[34–36], nucleic acid synthesis/amplification[37], cell-to-cell interaction assays[38], and material screening applications[39].

The minimum operational voltage of NOVAsort is also up to 30-fold lower, where at a throughput of 10 Hz only 20 $V_{pp}$ is needed for NOVAsort, in contrast to the 600 $V_{pp}$ requirement of a linear sorter. Supplementary Fig. 5 shows the strength of the electric field generated by the IDE and liquid 3D metal electrode, where the IDE-generated electric field is uniformly distributed on the surface area, while the electric field of the liquid metal electrode from the linear sorter decays along the Y direction. Even though a much lower voltage is applied, the strength of the electric field generated by the IDE is 3.3 times higher than that of the liquid metal electrode when compared with the region with the highest strength in the fluidic channel.

One aspect to note is that $Si_3N_4$ coating was applied on top of the IDEs to prevent damage to the electrodes during long-term operation of the droplet sorter. However, as the thickness of this dielectric coating layer increases, the required operational voltage also needs to be higher (Supplementary Fig. 12). The required voltage when no dielectric coating was applied (0 nm) was 23.4 times lower than that of the thickest layer (500 nm) tested. Their relationship is not linearly correlated because the strength of the electrical field diminishes exponentially as the distance increases. A device with a 500 nm coating has been successfully tested with no malfunctioning even after 3 days

of continuous operation at a throughput of 30 Hz. Thus, balancing the operational voltage requirement and the longevity of the device may have to be made depending on the application need, which may determine the thickness of the desired dielectric layer.

Balancing hydrodynamic resistance is critical during any droplet microfluidics operation. Just like conventional microfluidic droplet sorters, NOVAsort needs hydrodynamic resistance of all three outlets to be well balanced for optimal performance. In the absence of any input voltage, all droplets should exit from the waste 2 outlet. An exception to this rule is extremely large droplets that naturally exit through the waste 1 outlet because of pinched flow fractionation. For all experiments reported here, the outlets were balanced by adjusting the heights of the collection chambers for each outlet. It is worth noting that the ratio of flow rate of the hit outlet vs that of the waste 2 outlet determines the position of the critical border (Fig. 4H). The ratio of flow rates can be modified by either changing the input oil flow rate or the outlet channel resistances. The flow rate (hit outlet vs waste 2 outlet) ratio adopted in this paper is 0.9 (Supplementary Fig. 9). The detailed balancing method is elaborated in supplementary text and Supplementary Fig. 14.

The electric field of the linear sorter could cover a large region in the fluidic channel, therefore shielding electrodes are always needed to prevent unwanted merging of droplets outside of the sorting zone. Such shielding electrodes add significantly to the footprint of the device, as well as increase the fabrication steps and cost. Since the electric field generated by the IDEs is spatially localized and the applied voltage is one order of magnitude lower than that needed for a linear sorter, no shielding electrodes are needed. Moreover, unwanted droplets merging within the IDE region were not observed even if the droplets were in direct contact with each other (Fig. 4D and Supplementary Movie 6). The only observed case of unwanted droplet merging using the IDEs occurred when two requirements were met: (i) The IDE finger-to-finger distance is large enough, in this case, the IDE generated electric field can reach the contact area of two or more droplets; (ii) a high voltage applied to achieve sufficiently high electric field to reach the droplet-to-droplet contact area (200+ $V_{pp}$ needed in our case). A more detailed concept is elaborated in our previous publication[30]. The electric field generated by our IDEs having an electrode-to-electrode distance of 5 µm does not extend sufficiently from the electrode surface; therefore, merging is less likely to occur in the IDE design. The only failure mode of NOVAsort is when a small fluorescent droplet is close to or in contact with correctly sized waste droplets. In such a case, when the electric field is triggered, the waste droplet can be sorted as a false-positive case. This failure mode is shown in Supplementary Movie 9. In the current IDE design, >95% field strength reaches a height of 20 µm (Fig. 4). Therefore, if the channel height increases, target droplets might not be in range of the IDEs. Thus, the channel height needs to be tailored to the target size of interest, where the most important feature is the distance from the top of IDE fingers to the bottom of the target droplets. As to a very small target droplet size, as long as the droplet diameter is large enough to prevent unintentional coalescence and the channel height is low enough to bring the droplet within range of the IDEs, the IDEs can actuate the droplets without disruption.

The density difference between the aqueous and oil phases (buoyancy force) can impact the time needed for a droplet to float up and contact the channel ceiling. If the density difference is smaller, a longer time is needed before the droplets reach the sorting region, otherwise the sorting accuracy or throughput could be lower. To tackle this potential issue, higher-density carrier oil (e.g. FC40, density = 1.86 $g·ml^{-1}$) can be a substitute to maintain enough density difference. Also, another solution is to integrate a droplet lifting junction as shown in Section 1 (Fig. 2). The NOVAsort also can be implemented when the density of aqueous phase is higher than the oil phase (mineral oil for example). In this case, IDE patterns need to be

placed on the top of the channel rather than bottom, so that sinking droplets can be sorted.

In summary, NOVAsort allows the recovery of target-size, fluorescence-positive droplets from either an ideal monodisperse or a highly polydisperse droplet library. Even in the case of high input droplet polydispersity, the accuracy of the system was greater than 99% at throughputs up to 235 Hz. The sorter also shows good tolerance to insufficient droplet-to-droplet spacing. The accuracy remained high (96.6%) even when the spacing was very small (200 μm). Additionally, unintentional droplet coalescence was rarely observed even when droplets made contact each other in the sorting section. This device shows minimal droplet disruption even at a relatively large droplet size (110 μm) and high through (100 Hz). The NOVAsort was also evaluated and compared with a conventional droplet sorter in an IVTT mock screening workflow, where NOVAsort improved the efficiency of the IVTT assay by 217%. The simple microstructure and standard fabrication protocol allow NOVAsort to be easily integrated into a broad range of droplet microfluidics-based screening applications.

## Methods

### Device fabrication
The fluidic channels of the IDE sorter presented here require a smooth transition between regions having different channel heights. Traditional photolithography is not able to create Z-direction sloped structures. Thus, a two-photon polymerization instrument was used to fabricate the master mold having sloped structures for the microfluidic device (2PP; Nanoscribe Photonics Professional GT, IP-Q photoresist)[40]. Replicas of the device were created in polydimethyl siloxane (PDMS) using conventional soft lithography techniques. IDE patterns (Ti/Au; 20 nm/200 nm) were created using standard microfabrication methods (E-beam evaporation of thin film metal followed by photolithography and metal etching). To protect the contents of the aqueous droplets from direct contact with the IDE layer, a passivation layer of silicon nitride ($Si_3N_4$, 500 nm thickness) was deposited using plasma-enhanced chemical vapor deposition (Oxford PlasmaLab 80 plus). The PDMS fluidic layer and the SiN-coated IDE-patterned glass slides were then treated with oxygen plasma for surface activation to enhance bonding. The fluidic layer was aligned with the IDE patterns and then bonded together at 85 °C for 8 h. Immediately before experiments, the fabricated device was treated with fresh, filtered Aquapel (Pittsburgh Glass Works LLC., USA) to make the internal surfaces of the microfluidic channel hydrophobic. The device was then rinsed using NOVEC 7500 oil (3 M) to remove any residual Aquapel. A schematic of the entire fabrication process is detailed in Supplementary Fig. 13.

As to the linear sorter, the PDMS layer design was fabricated by conventional photolithography (see supplementary document for detail) and the microfluidic electrode channel was filled with Field's metal (RotoMetals, Inc., San Leandro, CA), producing a three-dimensional electrode[15].

### Proof-of-concept and sorting accuracy testing
Six distinct populations of droplets were generated separately using flow-focusing droplet generators (120 μm-blank, 120 μm-bright, 95 μm-blank, 95 μm-bright, 50 μm-blank, and 50 μm-bright; all dimensions are in diameter). The "bright" droplets contained 8 μM DMAO (Biotium, CA, USA) and 8 μM SBADA (TOCRIS, MN, USA), along with black ink diluted in distilled water, while the "blank" droplets only contained distilled water. Therefore the "bright" droplets appear brighter in the green fluorescence channel, but darker in the brightfield images/videos. NOVEC 7500 oil (3 M) with 2% Picosurf-1 (Sphere Fluidics) was used as the carrier oil. All 6 droplet populations were mixed into a 6-droplet pool, which was reflowed into the sorting devices at a flow rate of 35 μL·h⁻¹. This resulted in a throughput of 30 Hz. The spacing flow rate was 500 μL·h⁻¹, and bias 1 and bias 2 were

set to 1200 μL·h⁻¹ and 1800 μL·h⁻¹, respectively. Syringe pumps were used to control the flow rates (KD Scientific Legato series). A 10 kHz, 60 $V_{pp}$ square wave was applied to the first set of IDEs using a waveform generator (Rigol DG4102) to actuate the large droplets. Bright droplets passing through the detection region would trigger the second set of IDEs with a 10 kHz, 40 $V_{pp}$ square wave for a duration of 40 ms through a feedback loop. The peak-to-peak voltage ($V_{pp}$) and sorting duration were increased depending on the droplet throughput. A 60 frame per sec (fps) video of all three outlets was acquired (Hamamatsu C1440) and analyzed to demonstrate the sorter's performance.

For the next sets of quantitative sorter characterization, a highly polydisperse "blank" droplet library was prepared by sonicating distilled water in carrier oil (Ultrasonic Cleaner, Branson 2800) for 10 min. The 95 μm diameter bright droplet population described above was added to this library at a 1:10 (v:v) ratio. This pool of droplets was first imaged and then reflowed into the sorting devices. Flow rates, throughput, and electric signal conditions for this experiment were the same as the ones described earlier. The output of each outlet was collected and imaged. High frame rate videos (Hamamatsu C1440) were acquired during both experiments to visualize the device performance as well as to identify failure modes. The same experiment was repeated for both NOVAsort and conventional linear droplet sorter.

### Device performance characterization
**Droplet throughput vs. applied voltage.** A monodisperse droplet library was prepared by adding 95 μm droplets containing 8 μM DMAO and 8 μM SBADA to 95 μm droplets containing distilled water at a ratio of 1:10 (v: v). This library was reflowed into the IDE sorter device at flow rates of 11 Hz, 34 Hz, 110 Hz, and 250 Hz. The flow rates were scaled up until adequate spacing was observed between consecutive droplets. A high frame rate video was acquired after letting the device stabilize for 5 min, and the droplet sorting duration was set based on how long it took for each droplet to traverse the width of the sorting IDE. The sorting voltage was increased in increments of 10 $V_{pp}$ from a starting voltage of 40 $V_{pp}$. The lowest value at which the device was able to efficiently sort droplets was recorded. The same experiment was repeated using a conventional linear sorter.

**Droplet library composition vs. sorting accuracy.** Two monodisperse droplet libraries were prepared that contained either 5% bright (95% blank) or 95% bright (5% blank) droplets. Each library was processed through NOVAsort and the conventional linear sorter at a throughput of 30 Hz. The output of each device was collected, imaged, and analyzed to determine the sorting accuracy.

**Passivation layer thickness vs. applied voltage.** Multiple devices were prepared with varying thicknesses of silicon nitride. A monodisperse library containing 10% fluorescent droplets having diameters of 90 μm was reflowed through each device at 30 Hz. The lowest voltage that achieved high (>99%) sorting efficiency was recorded.

### Image acquisition and analysis
Droplets collected from each experiment were reinjected into a droplet imaging chamber composed of pillars with 20 μm spacing for holding the collected droplets but allowing carrier oil to flow through (Supplementary Fig. 14). A Zeiss Axio Observer inverted fluorescence microscope was used to acquire tiled images of the droplets in both brightfield and GFP channels. Each tile of the larger image was exported as a separate TIFF file. An image analysis pipeline was developed in MATLAB to quantify the size of observable droplets in each image. Briefly, locations and radii of droplets within various size ranges were extracted using MATLAB's implementation of a two-stage Hough circle transform[10]. The radius search ranges deliberately contained overlaps to avoid blind spots in droplet detection. This resulted

in duplicate measurements, which were identified using a K-nearest neighbor analysis. Droplets less than 1 radius away from their nearest neighbor were excluded from the dataset. A scaling factor of 0.65 µm·pixel$^{-1}$ was determined for the imaging setup and was used to convert the measurement units from pixels to µm. The location and radius of each droplet were converted into a binary image mask and applied to the fluorescence image of the relevant tile. The mean grayscale intensity of the pixels within the mask was recorded. Droplets smaller than 10 pixels in radius, and those intersecting the edge of the tiles, were excluded from the analysis.

### Droplet size and brightness quantification from videos
Frames of each video were cropped to isolate the observation chambers before each outlet ("OB1/2/3" in Fig. 2C). The same pipeline described in the previous section was applied to each frame. The units of measurement were converted from pixels to micrometers using scaling information from the microscope being used. A frame containing no droplets from each outlet was blurred using a Gaussian filter with a radius of 50 px. The mean intensity of the resulting image was subtracted from the measured intensity of each droplet from the outlets to alleviate the effects of vignetting on the intensity measurements from each outlet.

### In-droplet IVTT workflow
IVTT of GFP was carried out in droplets to demonstrate the utility of the developed NOVAsort. The plasmid pJL1-sfGFP was a gift from M. Jewett (Addgene plasmid no. 102634). The plasmid was extracted using the ZymoPURE plasmid miniprep kit (Zymo Research, CA, USA). Forward primer 5′ CGAAATTAATACGACTCACTATAG 3′ and reverse primer 5′ TTCTAATCAGAATTGGCTTTCAGC 3′ were used to amplify the GFP sequence to generate the DNA template for the New England Biolabs (NEB) PURExpress in vitro protein synthesis reagent (NEB, #E6800L). Droplet generation (50 µm diameter) and droplet PCR amplification were carried out using the protocol described by Sukovich et al.[41]. To carry out IVTT of GFP, PCR-amplified droplets were reflown and merged with droplets (80 µm in diameter) containing the NEB PURExpress protein synthesis kit, which included the NEB PURExpress solution A (40% v/v), NEB PURExpress solution B (30%), RNAsin ribonuclease inhibitor (4%), PURExpress Disulfide Bond Enhancer 1 (2%), PURExpress Disulfide Bond Enhancer 2 (2%), and water (22%). The merged IVTT droplets were collected in a syringe and incubated at 37 °C for 6 h. The fluorescence intensity of the expressed GFP was measured at an excitation wavelength of 479 nm and an emission wavelength of 520 nm. The PCR products generated using high-fidelity Q5 polymerase were cleaned and concentrated using the Zymo Research DNA clean and concentration kit and used as a positive control.

### Statistics & reproducibility
No statistical method was used to predetermine the sample size. In certain experiments, droplets under 20 µm in diameter were excluded from the analyses. The experiments were not randomized. The Investigators were not blinded to allocation during experiments and outcome assessment.

### Reporting summary
Further information on research design is available in the Nature Portfolio Reporting Summary linked to this article.

## Data availability
The data generated for this study has been deposited in a Figshare repository (https://doi.org/10.6084/m9.figshare.26811331). Further data supporting the findings of this study are provided in the Supplementary Information. Source data is provided in the online version of this paper. Source data are provided with this paper.

## Code availability
Custom code developed for this study is also available via the FigShare repository (https://doi.org/10.6084/m9.figshare.26811331).

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

## Acknowledgements

The project depicted was sponsored by the Defense Advanced Research Projects Agency cooperative agreement W911NF1920013, Army Research Office agreement W911NF-19-1-0290, and Army Research Laboratory agreement W911NF-17-2-0144. The project was also supported by the National Institutes of Health / National Institute of Allergy and Infectious Diseases grants R01AI141607, R01AI168685, and R21AI139738. The content of the information does not necessarily reflect the position or the policy of the government, and no official endorsement should be inferred.

## Author contributions

H.Z.: Conceptualization, Methodology, Validation, Investigation, Writing—original draft, Writing—review & editing, Data curation. R.G.: Conceptualization, Methodology, Validation, Investigation, Writing—original draft, Writing—review & editing, Data curation. Y.L.: Data curation. Writing—original draft. C.H.: Conceptualization, Writing—review & editing, Data curation. A.G.: Conceptualization, Writing—review & editing. J.J.H.: Data curation. Haemin Jung: Data curation. R.S.: Data curation. P.deF.: Resources, Writing—review & editing, Supervision, Funding acquisition. A.H.: Conceptualization, Resources, Writing—review & editing, Supervision, Funding acquisition.

## Competing interests

The following patent was filed by Texas A&M University in January 2022: A.H., H.Z., C.H., A.G., J.D., and R.G. Interdigitated Electrode-Based Droplet Manipulation in Microfluidic Systems, U.S. Provisional Patent, no. 63293,812. All other authors declare that they have no competing interests.
