## [Peer Review File · Nature Communications]

Reviewers' Comments:

Reviewer #1:

Remarks to the Author:

This manuscript introduces a droplet sorter called NOVA sorter, which sorts droplets based on their size and fluorescent intensity. The sorter utilizes two Interdigitated Electrode (IDE) arrays to generate a localized electric field at the bottom of the channel, enabling it to divert droplets of specific sizes. Smaller droplets are not influenced by the electric field and continue to flow towards the waste outlet. The IDE arrays, along with the fluidic channel of varying heights, function as low-pass and high-pass size-based droplet filters. When combined with a fluorescence-activated droplet sorter, NOVA sorter can selectively sort target droplets that are both the correct size and fluorescent positive from an extremely polydisperse library, achieving high accuracy (>99%) at a throughput of up to 235 Hz. The use of IDEs addresses the failure modes of existing linear sorters and prevents droplet splitting during sorting. Compared to conventional linear sorters, NOVA sorter outperforms them in all aspects such as negative sorting accuracy. Additionally, the efficacy of the NOVA sorter was demonstrated successfully using an IVTT (in vitro transcription and translation) workflow, resulting in a significant improvement in sorting accuracy.

The manuscript is well-written, and the hypothesis is supported by extensive experimentation and data analysis. Although many of the images used in the manuscript show a small number of droplets for each experiment, the videos clearly demonstrate the use of large number of droplets and the accuracy of the experiment and leave no doubt that this sorter outperforms conventional linear sorters.

I find the experiments performed, data analysis, and discussion satisfactory. The device design is novel too. I did not observe any weaknesses in the experiments or results. Therefore, I strongly support publishing the manuscript as is.

Reviewer #2:

Remarks to the Author:

The authors proposed a novel accurate droplet sorter based on opto-volume, namely NOVA sort, which fulfilled the highly precision in the six diverse conditions. Such as the devices allow to collect target-sized fluorescent-positive hit droplets from either an ideal monodisperse or an extremely polydisperse droplet composition. The operation accuracy of the sorting system reached more than 99% with 235Hz and the lower driving voltage of 20 Vpp. Therefore, the research results are noteworthy and support the conclusions and claims. The methodology sounds good. So I agree with publishing on

Nature Communication after language editing.

Reviewer #3:

Remarks to the Author:

The manuscript discusses the development a new droplet-sorting platform (NOVAsort) that allows to discern droplets based on both size and fluorescence intensity. The study demonstrates that utilizing two-step process in which the droplet population is prescreened via passive, size-based separation followed by active, fluorescence and size-based separation, allows to collect target-sized fluorescent-positive hit droplets with unprecedented accuracy. The NOVAsort droplet sorter makes uses of surface IDE-generated DEP force and also droplet buoyancy phenomena to achieve excellent tolerance to polydispersity of input droplet size at a relatively small droplet-to-droplet distance and low operational voltage. The authors also showcase the use of NOVAsort platform in IVTT screening application, compared to conventional droplet sorting technologies. The topic of this study in enhancing the practicality of microfluidic high-throughput screening is interesting and would attract attentions from a large audience. Overall, the results of this work are useful and provide remarkable progresses in the field. Therefore, I recommend consideration of this work in Nature Communications, and hope that the following minor suggestions would further improve the work.

1. Although the platform presented seems highly efficient in reducing false positives and false negatives, the throughput is not thoroughly compared with other state-of-the-art technologies. It might be better to discuss on this aspect with platforms other than a linear sorter. In other words, is 30 droplet/sec sufficient for all the applications proposed?
2. The description in page 4 that first introduces the sorting mechanism is difficult to follow through. Isn't the buoyant force applied to larger droplets larger than the smaller analogues? If the primary reason that larger droplets selectively experience the localized electric field from the IDEs located at the bottom of the channel, it might be better to redraw the subfigure shown in Figure 1D as it is confusing. One suggestion is to revise Figure 1D similar to how it is schematically illustrated in Figure 5A which more explicitly describes the flow field and the forces imposed on the droplet.
3. In Figure 2A Section 3, is the legend correct? The manuscript mentions 120 micron in height but the legend is written as 160 micron for blue region.
4. How effective is NOVAsort in separating "Disrupted/split, Fluor +" if their size is larger and closer in size to the hits?
5. In page 7, line 254, typo, "NOVA sort provides (10?)X better..."
6. One of the key design considerations in a droplet sorter is the IDE pattern design and their compatibility with the droplet size and the channel dimensions. Any discussion on

this would be beneficial for readers who are interested in such platforms.

7. Does the density of the dispersed droplet and the viscosity affect the sorting performance?

Comments from the reviewers:

Reviewer 1

No questions or comments need to be answered from this reviewer.

Reviewer 2

The authors proposed a novel accurate droplet sorter based on opto-volume, namely NOVA sort, which fulfilled the highly precision in the six diverse conditions. Such as the devices allow to collect target-sized fluorescent-positive hit droplets from either an ideal monodisperse or an extremely polydisperse droplet composition. The operation accuracy of the sorting system reached more than 99% with 235Hz and the lower driving voltage of 20 Vpp. Therefore, the research results are noteworthy and support the conclusions and claims. The methodology sounds good. So I agree with publishing on Nature Communication after language editing.

Answer: We have thoroughly gone through the manuscript and completed comprehensive language editing. We have corrected several typos, grammar mistakes, and presentation errors in the figures.

Reviewer 3

1. Although the platform presented seems highly efficient in reducing false positives and false negatives, the throughput is not thoroughly compared with other state-of-the-art technologies. It might be better to discuss on this aspect with platforms other than a linear sorter. In other words, is 30 droplet/sec sufficient for all the applications proposed?

Answer: The highest throughput we reported in this paper is 235 Hz. We tested up to this throughput based on the maximum frame rate (~500 FPS at cropped frame) of our high-speed camera system that can provide error-free analysis of droplets. However, the throughput limit of the NOVA sort system can be much higher than 235Hz, based on image analyses of output droplets (although this does not allow in-depth analyses of analyzing video frame by frame). In addition, for polydisperse droplet library screening, we do not count droplets that are less than 20 μm for throughput calculation, thus the actual throughput should be much higher. While several higher throughput (>1000 droplets/s) platforms have been reported for similar droplet sizes [droplet diameter of 100 μm , DOI: 10.1126/sciadv.aba6712], their performance has not been evaluated against polydisperse droplet libraries. This is likely because their performance is expected to suffer under these circumstances. This discussion has been added to the manuscript (highlighted in red, page 16).

A throughput of 235 Hz allows NOVA sort to process >800K droplets/h, which is adequate for most biological applications (the typical library sizes to be screened are ~1M library size). NOVA sort can process 1M samples in about 1.25 hours. Examples of published works using similar throughputs and/or library sizes are below:
<https://doi.org/10.1038/s41587-020-0466-7>,
<https://doi.org/10.1073/pnas.1204514109>.

This discussion has now been added to the manuscript (highlighted in red, page 16).

2. The description in page 4 that first introduces the sorting mechanism is difficult to follow through. Isn't the buoyant force applied to larger droplets larger than the smaller analogues? If the primary reason that larger droplets selectively experience the localized electric field from the IDEs located at the bottom of the channel, it might be better to redraw the subfigure shown in Figure 1D as it is confusing. One suggestion is to revise Figure 1D similar to how it is schematically illustrated in Figure 5A which more explicitly describes the flow field and the forces imposed on the droplet.

Answer: The description in Figure 1D was meant to show that all droplets experience a sufficiently large buoyancy force to push them up to the ceiling. Beyond that, the difference between the sizes does not matter. All droplets being buoyant keep larger droplets within range of the IDEs patterned at the bottom of the channels, while smaller droplets that are also buoyant are further away from the IDE patterns on the surface of the channel. Figure 1D has been redrawn to clarify this point.

3. In Figure 2A Section 3, is the legend correct? The manuscript mentions 120 micron in height but the legend is written as 160 micron for blue region.

Answer: This typo and label errors in Figure 2 have been resolved.

4. How effective is NOVA sort in separating "Disrupted/split, Fluor +" if their size is larger and closer in size to the hits?

Answer: Merged/Split droplets are rarely similar to the original target droplet size. The IDE design we are currently using has about 6-7 μm separation resolution in terms of droplet diameter. Please see our previous result (DOI: 10.1126/sciadv.abc9108), which we have referenced in our discussion section. This means if the disrupted/split droplets are within 6-7 μm of our target droplets' diameter, even though that is unlikely, the system will consider the droplets to be true hits, even if they are not. Higher resolution can be achieved by reducing the IDE electrode-to-electrode distance. A downside of this is that having a higher resolution distance between the electrodes will be more challenging for microfabrication.

5. In page 7, line 254, typo, "NOVA sort provides (10?)X better..."

Answer: The problem has been fixed. "NOVA sort provides approximately 10X better".

6. One of the key design considerations in a droplet sorter is the IDE pattern design and their compatibility with the droplet size and the channel dimensions. Any discussion on this would be beneficial for readers who are interested in such platforms.

Answer: We agree with the reviewer's comment that such an aspect is important, and thus we have included the following in the "Discussion" section.

First, IDEs need to be wide enough to fill the channel width to capture droplets on time during fluorescence-activated droplet sorting. Second, we have shown here that the IDE pattern used here is compatible with droplets in the 50-120 μm diameter range. In the current IDE design, 95% of the electric field strength reaches a height of 20 μm (Figure 4, and supplementary document

of DOI: 10.1126/sciadv.abc9108). Therefore, if the channel height increases, the target droplets may not be in the range of the IDEs, and the channel height must be tailored to the target size. The most important feature is the distance from the bottom of the channel/top of IDE fingers to the bottom of the target droplets. As to a very small target droplet size, as long as the droplet diameter is large enough to prevent unintentional droplet coalescence and the channel height is low enough to bring the droplet within range of the IDEs (See Figure S8 in DOI: 10.1126/sciadv.abc9108), the IDEs will be able to actuate the droplets without disruption. Unintentional droplet coalescence in this context is caused by the electric field being strong enough at the contact point of two adjacent droplets to cause depolarization of the surfactant (the mechanism has been explained in the discussion section, page 18). The updated discussion is highlighted in red (Page 18).

7. Does the density of the dispersed droplet and the viscosity affect the sorting performance?

Answer: The density of the droplet itself does not affect the droplet sorting performance. However, the density difference between aqueous phase and oil phase, namely buoyancy, can impact the time needed for a droplet to float up and contact the channel ceiling. At a very high flow speed, a smaller density difference will require longer time for the droplet to float up and thus could reduce the sorting accuracy. To tackle this issue, higher-density carrier oil (e.g. FC40, density = 1.86 g/ml) can be used to maintain enough density difference. Also, this is the major reason we integrated a droplet lifting junction in section 1 (Figure 2) to help with droplets floating up. In most biological applications, the aqueous phase consists of cell culture media, buffer solution, or hydrogel, whose densities are within 1.0 to 1.2 g/ml. The NOVEC 7500 carrier oil used in this study has a density of 1.614 g/ml, which can provide sufficient buoyancy/density difference (this information has been updated in the discussion part of the manuscript, highlighted in red, Page 18).

The developed method can also be used in the case where the density of carrier oil is lower than that of the aqueous solution (this had been already discussed in the manuscript, page 18).

We speculate that the major impact of high-viscosity droplets will be that the shape/circularity (length/width) of the droplets may be impacted, primarily caused by shear force and imperfect hydrophobic surface treatment. In turn, this can affect the system resolution and the passband of the size-based separation method. However, at this point we have no data to support this hypothesis as we have not tested the device using aqueous solution of different viscosity. This can be further tested in our future work.

Reviewers' Comments:

Reviewer #3:

Remarks to the Author:

The authors have addressed the concerns and issues raised. Therefore, I recommend publication with no further revision.